# Cor Triatriatum Dexter: Contrast Echocardiography Is Key to the Diagnosis of a Rare but Treatable Cause of Neonatal Persistent Cyanosis

**DOI:** 10.3390/children9050676

**Published:** 2022-05-06

**Authors:** Irene Picciolli, Gaia Francescato, Anna Maria Colli, Alessia Cappelleri, Alessandra Mayer, Roberto Raschetti, Roberta Di Cosola, Marco Pisaniello, Giuseppe Alberto Annoni, Marco Papa, Mimoza Maldi, Guido Olivieri, Fabio Mosca, Stefano Marianeschi

**Affiliations:** 1Neonatal Intensive Care Unit, Fondazione IRCCS Ca’ Granda Ospedale Maggiore Policlinico, 20122 Milan, Italy; irene.picciolli@policlinico.mi.it (I.P.); alessia.cappelleri@policlinico.mi.it (A.C.); alessandra.mayer@policlinico.mi.it (A.M.); roberto.raschetti@policlinico.mi.it (R.R.); fabio.mosca@policlinico.mi.it (F.M.); 2Pediatric Cardiology, Fondazione IRCCS Ca’ Granda Ospedale Maggiore Policlinico, 20122 Milan, Italy; anna.colli@policlinico.mi.it (A.M.C.); roberta.dicosola@policlinico.mi.it (R.D.C.); marco.pisaniello@policlinico.mi.it (M.P.); 3Pediatric Cardiology, ASST Grande Ospedale Metropolitano Niguarda, 20162 Milan, Italy; giuseppealberto.annoni@ospedaleniguarda.it (G.A.A.); marco.papa@ospedaleniguarda.it (M.P.); 4Congenital Cardiac Surgery, ASST Grande Ospedale Metropolitano Niguarda, 20126 Milan, Italy; mimoza.maldi@ospedaleniguarda.it (M.M.); guidomaria.olivieri@ospedaleniguarda.it (G.O.); stefano.marianeschi@ospedaleniguarda.it (S.M.)

**Keywords:** neonatal persistent cyanosis, cor triatriatum dexter, right atrial membrane, echocardiography

## Abstract

Cor triatriatum dexter (CTD) is an extremely uncommon and underreported congenital cardiac anomaly in which the persistence of the embryonic right venous valve separates the right atrium into two chambers with varying degrees of obstruction to antegrade flow and variable degree of right to left shunt at atrial level. Depending on the size of the valves, clinical manifestations vary from absence of symptoms to severe hypoxia, requiring urgent surgical correction. We herein describe the diagnostic difficulties in a case of neonatal CTD, who developed increasingly severe and unresponsive cyanosis, first interpreted as postnatal maladjustment with pulmonary hypertension. The failure to respond to oxygen and pulmonary vasodilators led us to reconsider a different diagnosis. The use of contrast echocardiography improved the diagnostic performance of transthoracic echocardiogram (TTE) and revealed a massive right-to-left shunt secondary to the presence of an atrial membrane that required urgent surgery.

## 1. Introduction

Cor triatriatum dexter (CTD) is a very uncommon congenital heart disease (CHD) that derives from the persistence of the right venous valve, one of two valves that guards the sinoatrial orifice from the primitive atrium. The embryonic sinus venosus presents two horns: the right and the left horn. The right horn forms a major part of the right atrium and it is separated from the primitive atrium by the sinoatrial orifice. The sinoatrial orifice is flanked on each side by a valvular fold, the right and left venous valves. The right valve normally regresses between weeks 9 and 15 of gestation, leaving the crista terminalis superiorly and the Eustachian and Thebesian valves guarding the IVC and coronary sinus orifices, respectively. The persistence of the right venous valve results in partitioning the right atrium into two separate chambers (one smooth walled and the other trabeculated) and creates what is known as cor triatriatum dexter [1,2]. The smooth atrial portion receives venous blood from inferior vena cava, superior vena cava, and coronary sinus, while the trabeculated portion contains the right atrial appendage and the opening of the tricuspid valve.

CTD may be isolated or associated with other CHDs, such as atrial septal defects (ASD), pulmonary valve stenosis, tricuspid valve regurgitation and right ventricular hypoplasia. Clinical presentation differs widely, from an incidental finding to severe cyanosis and heart failure, depending on the degree of obstruction to antegrade flow [1,2]. Diagnosis is not always straightforward.

## 2. Case Report

We report the case of an infant born by spontaneous labor after an uncomplicated term pregnancy. Birth weight was 3950 gr (91° percentile). The infant was admitted to the neonatal intensive care unit (NICU) at 12 h of age: he presented with isolated cyanotic episodes with peripheral oxygen desaturation reaching 70%, only partially responsive to oxygen supplementation. There was neither respiratory distress (however mild) nor signs of early neonatal sepsis.

Chest X-ray was unremarkable except for the incidental finding of a right clavicular fracture. The baby therefore underwent a full cardiac investigation. Electrocardiogram was unremarkable. Trans-thoracic echocardiogram (TTE) showed situs solitus, levocardia with normal systemic and pulmonary venous returns, along with atrio-ventricular and ventriculo-arterial concordance. Although showing all correct components and a normally opening pulmonary valve, the right-sided chambers appeared somewhat small for a newborn still in a transitional stage. In the right atrium (RA), there was a membrane arising from the inferior vena cava (IVC), much like a Eustachian valve, but it appeared both to reach the interatrial septum and, being very mobile, to intermittently engage through the tricuspid valve orifice (Figure 1, and Appendix A). Tricuspid antegrade flow was clearly demonstrated, and therefore the membranous structure was initially interpreted as a redundant Eustachian valve and the hypoxia attributed to postnatal maladjustment.

During the first few days of life, peripheral oxygen saturation (SpO_2_) was 92–95%, but with oxygen desaturation (SpO_2_ < 80%) in the course of routine neonatal care and meals. Due to worsening, albeit still mostly intermittent, hypoxia respiratory support with nasal CPAP together with inhaled nitric oxide was started. FiO_2_ was initially 0.40 and progressively increased up to 0.50. On day 5 of life, additional oral pulmonary vasodilators (sildenafil) were started at a dose of 2 mg/kg 4 times a day with apparent clinical improvement, which proved to be transient. As effective response failed to be stable, inhaled nitric oxide was discontinued after 7 days. On day 11 of life, the infant developed sustained severe hypoxia with SpO_2_ 75–85% in nasal CPAP with FiO_2_ of 1.0. Two-dimensional echocardiography remained unchanged, failing to provide an explanation. We therefore decided to proceed to a contrast echocardiography via a peripheral line in the right foot. Informed consent for the procedure was obtained from the parents. We chose to inject a solution of agitated normal saline to produce microbubbles, as no commercial ultrasound-enhancing agents were available, and considering this to be safer in a newborn. Microbubbles streamed preferentially to the left atrium via the 4 mm foramen ovale, directed there by the membranous structure, which was becoming thicker and appeared to divide at least partially the atrium in two portions, although antegrade flow through the right ventricle and pulmonary artery was still well-represented.

Because of the small size of the foramen, which was at risk of becoming restrictive, the baby was referred for urgent surgical repair. After full sternotomy and cardio-pulmonary bypass were established, cold blood cardioplegia was delivered to arrest the heart. At right atriotomy, a membrane was found arising from the septal cusp of the tricuspid valve and the coronary sinus, extended to just above the IVC, obstructing the blood flow of tricuspid valve. The membrane was easily resected and the foramen ovale (FO) closed with a direct suture (Figure 2). The patient was weaned from bypass and the sternum closed as routine.

Postoperative transthoracic echocardiogram (TTE) confirmed the complete removal of the membrane in RA and the closure of the FO with no residual shunt and good biventricular function.

The postoperative course was uneventful. The patient was extubated on postoperative day (POD) 2, transferred from the Intensive Care Unit to the Neonatal Ward on POD 3 and discharged from hospital on POD 7. The baby is now almost 4 months old; he is completely asymptomatic with a normal cardiac physical examination and normal peripheral oxygen saturation. Electrocardiogram is normal, 2D echocardiography shows no images suggestive of residual membrane, the interatrial septum is intact and no pericardial effusion is present.

## 3. Discussion

Cor triatriatum dexter is an extremely rare malformation; its incidence is estimated to be below 0.01% of all CHD.

The main clinical presentation is neonatal central cyanosis: most systemic venous blood is directed to the left atrium through the patent foramen ovale. Different factors have been considered to impact the time of onset and severity of clinical manifestations: the obstructive versus non-obstructive nature of the membrane, which determines the degree of blood flow obstruction from the smooth sinus part of the RA toward its trabeculated part; the presence of associated right-sided heart malformations; the presence of functional tricuspid valve obstruction due to the protrusion of the membrane through the tricuspid orifice; and the presence of tricuspid valve insufficiency due to the traumatic effects of the membrane on its leaflets. Furthermore, the RA membrane could sometimes create an obstruction to inferior vena cava flow, especially when the foramen ovale is restrictive. The diagnosis of CTD can be extremely challenging and it is widely accepted that this anomaly is underreported. Even in the presence of neonatal cyanosis refractory to therapies, as in our case, this rare malformation may be not immediately recognized.

A very limited number of cases have been reported. In a recent review, Kalangos et al. [2] identified only 14 cases including their own. More often, cor triatriatum dexter is associated with additional right-sided heart malformation such as Ebstein’s anomaly, tricuspid or pulmonary valve stenosis, atresia or right ventricular hypoplasia. The first cases described have been incidental postmortem findings and presented complex congenital cardiac malformations. Four pediatric cases with isolated CTD presented in the first days or months of life with cyanosis and they were alive after surgery [2]. Another case, herein described by Galli et al., was quite similar: it was the case of an infant with central cyanosis in the second day of life, treated with oxygen supplementation, inhaled nitric oxide and oral sildenafil without improvement in arterial saturation [3].

Some adult milder cases refer to patients that remain asymptomatic or rarely develop signs of right-side obstruction in time [4].

Differential diagnosis using ultrasound is to be made with the much more common prominent Eustachian valve or redundant Chiari’s net, especially when the membrane is thin and mobile and engaging into the tricuspid valve orifice as in our case. Moreover, right to left shunt due to unfavorable streaming may not be immediately obvious, and initial response to oxygen administration can be deceiving. Therefore, if any doubt persists as to the nature of refractory cyanosis, contrast echocardiography, with bubbles contrast injected via a peripheral line connected to a lower limb, becomes mandatory.

Contrast echocardiography refers to diagnostic ultrasound of the heart, whereby acoustic enhancing agents, including agitated saline microbubbles, can transit to the systemic circulation after intravenous injection and can be detected in order to improve diagnostic performance. Injection of agitated saline has been used for decades to evaluate the presence of right-to-left intracardiac shunts or pulmonary arteriovenous malformation and continues to remain a valuable clinical tool in daily practice. Over the past decade, there has been a year-by-year increase in the performance of contrast echocardiography, and new ultrasound-enhancing agents (UEA) are available and characterized by viscoelastic properties designed to transit from the peripheral bloodstream into the right heart, through the pulmonary vasculature and into the left-sided chambers and systemic circulation. In 2019, the United States Food and Drug Administration (FDA) approved ultrasound-enhancing agents (UEA) for pediatric echocardiography. Contrast echocardiography has been established as safe and effective across a spectrum of pediatric clinical applications in small cohorts: it improves definition of ventricular anatomy and function especially in children with acoustic window limitations [5,6,7].

The use of contrast echocardiography in helping the diagnosis of cor triatriatum dexter was first described in adults. Modi K. et al. confirmed diagnosis of a rare variant of CTD in a 67-year-old man, where the transthoracic approach was limited even with the aid of color Doppler and where an invasive method of imaging such as transesophageal echocardiography (TEE) or cardiac MRI could not be employed [8]. In this case, the authors used an ultrasound-enhancing agent. More recently Theodoropoulos K.C. et al. described an unusual finding during contrast echocardiography with agitated saline performed to look for interatrial shunt in a 54-year-old man. The right atrium appeared divided in two compartments by a membrane; the compartment which was receiving blood from the superior vena cava was opacified, as the agitated saline was injected via an upper limb vein, while the compartment receiving blood from the IVC was only intermittently enhanced via the membrane fenestrations [9].

In our neonatal case, the usefulness of contrast echocardiography was focalized to detect and prove a right-to-left shunt directed by the membranous structure of the right atrium into the left atrium. The use of agitated saline microbubbles was sufficient for our purpose.

Moreover, in adults it is possible to use other diagnostic modalities to simplify diagnosis, such as transesophageal echocardiography (TEE), 3D echocardiogram or cardiac magnetic resonance (CMR). TEE is an optimal diagnostic modality to visualize atrial morphology and to better describe the atrial membrane, but in the neonatal population it is not universally available and most of the time transthoracic and subcostal approaches are more than adequate. Three-dimensional echocardiogram is not available in our NICU setting. Cardiac MRI is even less ubiquitous and a highly expensive investigation, undoubtedly useful in the adult setting when the transesophageal approach is not feasible for whatever reason. In infants, cardiac MRI is not the first choice due to the need for sedation, and in our setting it requires access to a different institution, transporting a child on ventilatory support. We would therefore deem this unnecessary if alternative reliable diagnostic tools are available on site, such as, in this specific case, contrast echocardiography.

Surgical treatment of CDH depends on the severity of clinical manifestations. It is necessary in cases of severe blood flow obstruction leading to central cyanosis, elevated central venous pressure and suprahepatic portal hypertension. Asymptomatic patients usually do not require any intervention. However, although rare, when the atrial septum is patent, individuals may present later in life with syncope, paradoxical systemic, and coronary embolism [4].

Reported operative mortalities vary from 0% to 29% in all case series. The higher mortalities are found in older series or in those with associated complex congenital heart diseases, presenting in infancy with cardiac failure or with associated severe pulmonary hypertension [10].

Treatment in infancy consists of resection of the abnormal tissue as well as repair of associated lesions, if any, while percutaneous attempts at resection have been described in symptomatic adult patients [11]. Alghamdi M. reported a 1-week-old baby with isolated cor triatriatrum dexter treated initially by cardiac catheterization but without success in disrupting the membrane [12]. In our case, the membrane was extremely thick so it is unlikely that percutaneous intervention would have succeeded.

In conclusion, our case shows the challenges of differential diagnosis of persistent hypoxia in the neonatal setting and the key value of bubble contrast echocardiography in allowing correct, timely and successful surgical treatment.

## Figures and Tables

**Figure 1 children-09-00676-f001:**
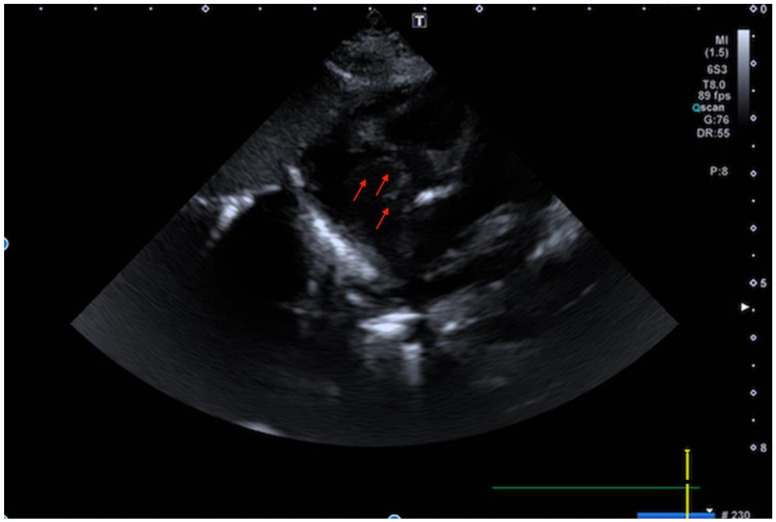
Transthoracic echocardiogram (TTE) (subcostal view) shows the membrane arising from the inferior vena cava (IVC) and reaching the interatrial septum.

**Figure 2 children-09-00676-f002:**
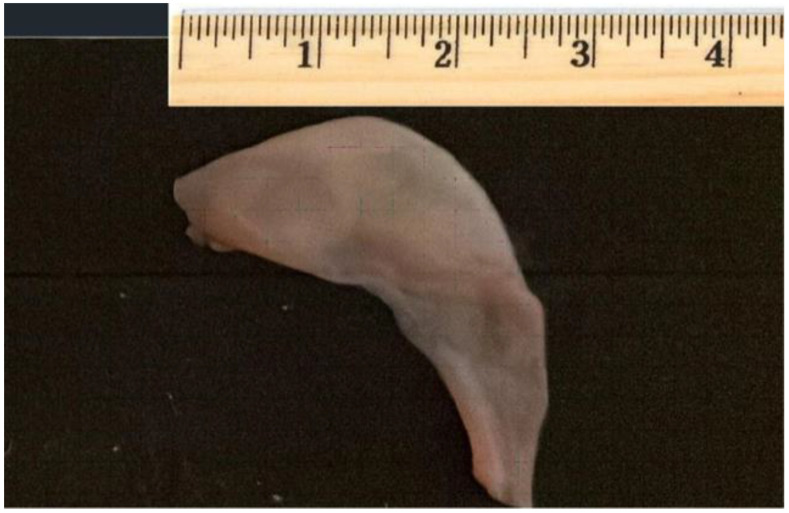
Membrane excised.

## Data Availability

No database is available due to the nature of the study (single patient observational case report).

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
