# Peer review of "Cor Triatriatum Dexter: Contrast Echocardiography Is Key to the Diagnosis of a Rare but Treatable Cause of Neonatal Persistent Cyanosis"

_children, 2022, doi:10.3390/children9050676_

Round 1

Reviewer 1 Report

Thank you for the opportunity to review this interesting case report, which is a description of how contrast echocardiography was used to diagnose cor triatriatum dexter (CTD) in a cyanotic neonate.  CTD is a rare cardiac malformation, described by others in the literature through several case reports along with reviews. The presentation of CTD can vary, ranging from asymptomatic to persistent neonatal cyanosis. The clinical presentation depends on the degree of obstruction caused by the membrane. Milder forms of CTD can go undiagnosed until late adulthood, and there is a need of considering other diagnostic methods than conventional echocardiography.

General comments:

The topic of this case report is interesting, but there are some major issues to be satisfactorily addressed before the paper could be considered suitable for publication.

  1. Contrast echocardiography is considered an extension of the conventional echocardiographic examination. It is not used in all echo-labs and therefore, this paper would benefit from a much more detailed description of contrast echocardiography, i.e. indications, side effects, costs etc.
  2. There are no references to contrast echocardiography provided, and there are several publications that could be used. Lindner et al published in 2021 a review of contrast echo, which is just one example (Lindner JR. Contrast echocardiography: current status and future directions. Heart. 2021 Jan;107(1):18-24. doi: 10.1136/heartjnl-2020-316662. Epub 2020 Oct 19. PMID: 33077502).
  3. Furthermore, contrast echo has been described earlier as a tool when diagnosing CTD in adults, which has not been mentioned. Suggestions are Theodoropoulos KC et al, Echocardiography. 2018 Nov;35(11):1895-1897 and Modi K et al, Echocardiography. 2009 Feb;26(2):220-3.
  4. The Discussion part should focus on the case and how it could be compared with previously reported studies or case reports. The first part of the discussion is far too broad and some of it would be better suited for the Introduction part. The readability of this section could be improved.
  5. The manuscript is written in an appropriate English, but a detailed spell check is required.

Specific comments:

Title:

  1. Typo: “…echocardiographys…”, should probably be echocardiography?

Abstract:

  1. Line 16 typo: “form” should probably be from? Furthermore, the whole sentence “We herein describe the diagnostic difficulties in a case of neonatal CTD, who developed increasingly severe and unresponsive cyanosis form massive right-to-left shunt and required urgent surgery.” is misleading as the actual diagnostic difficulties are poorly described. Please, rewrite or develop diagnostic difficulties in the text.

Introduction:

  1. As earlier mentioned, the beginning of Discussion would be better suited for Introduction.

Case report:

  1. At what age was the baby admitted to NICU, and what were the symptoms, only cyanosis?
  2. What do you mean by “lack of response for several days”? Please, describe what oxygen saturation levels the baby expressed during this time? What kind of respiratory support was used during this time?
  3. Do you have any more data on outcome after POD 7?

Figures:

  1. Fig 2A needs a higher resolution, but I would consider this figure to be left out, since the quality of the image is suboptimal and does not add any information of importance. Fig 1 and Fig 2B are appropriate.
  2. Videoclips 1-3 are showing presurgery images by conventional If there are any videoclips showing the contrast echocardiography examination, please add this file.

Discussion:

  1. As earlier mentioned, this part needs to be more of a discussion and not only a broad description of CTD.
  2. There is a lack of comparisons of contrast echocardiography to other diagnostic modalities e.g. TEE, 3D echo and cardiac MRI, when suggesting this as a key method for diagnosing CTD. Please, provide information on pros and cons when considering choice of method.

Author Response

Please see the attachment "Response to REVIEWER 1" .

We created a new version of the article revised with reviewer's suggestions.

Reviewer 2 Report

The authors report a newborn with cor triatriatum dexter, a rare anomaly. The patient was cyanotic and required resection of the obstructive membrane. The report is well written and is supplemented by images which illustrate the anatomy. Can the authors elaborate a bit more in the discussion section about indications for operative repair of this rare anomaly?

Round 2

Reviewer 1 Report

Response from reviewer 1 to authors:

Thank you for revising the manuscript sufficiently. The new parts, including a detailed description of the contrast echocardiography method, have definitely improved the content. There are still a few things to be considered before publication.

  1. There is a need of English language editing of the new parts.
  2. The Discussion part has been improved, but I suggest the authors to take this another round to reduce the text a little.
  3. The new references (e.g. ref 9, 10) have to be presented in a proper style according to author’s instructions.

Author Response

Point 1: There is a need of English language editing of the new parts.

 Response 1: We have hopefully removed any misspellings and typos and made an Engish language editing.   

Point 2: The Discussion part has been improved, but I suggest the authors to take this another round to reduce the text a little.

Response 2: The text has been reduced a little

 Point 3: The new references (e.g. ref 9,10) have to be presented in a proper syle according to author’s instructions.

 Response 3: All the refereces have been corrected in a proper style according to author’s instructions.